# Global prevalence and case fatality rate of Enterovirus D68 infections, a systematic review and meta-analysis

**Amary Fall**[1], **Sebastien Kenmoe**[2,3]*, **Jean Thierry Ebogo-Belobo**[4], **Donatien Serge Mbaga**[5], **Arnol Bowo-Ngandji**[5], **Joseph Rodrigue Foe-Essomba**[6], **Serges Tchatchouang**[7], **Marie Amougou Atsama**[8], **Jacqueline Félicité Yéngué**[9], **Raoul Kenfack-Momo**[10], **Alfloditte Flore Feudjio**[10], **Alex Durand Nka**[11], **Chris Andre Mbongue Mikangue**[5], **Jean Bosco Taya-Fokou**[5], **Jeannette Nina Magoudjou-Pekam**[10], **Efietngab Atembeh Noura**[4], **Cromwel Zemnou-Tepap**[10], **Dowbiss Meta-Djomsi**[8], **Martin Maïdadi-Foudi**[8], **Ginette Irma Kame-Ngasse**[4], **Inès Nyebe**[5], **Larissa Gertrude Djukouo**[10], **Landry Kengne Gounmadje**[10], **Dimitri Tchami Ngongang**[5], **Martin Gael Oyono**[9], **Cynthia Paola Demeni Emoh**[5], **Hervé Raoul Tazokong**[5], **Gadji Mahamat**[5], **Cyprien Kengne-Ndé**[12], **Serge Alain Sadeuh-Mba**[2], **Ndongo Dia**[1], **Giuseppina La Rosa**[13], **Lucy Ndip**[3], **Richard Njouom**[2]*

**1** Virology Department, Institute Pasteur of Dakar, Dakar, Senegal, **2** Virology Department, Centre Pasteur of Cameroon, Yaounde, Cameroon, **3** Department of Microbiology and Parasitology, University of Buea, Buea, Cameroon, **4** Medical Research Centre, Institute of Medical Research and Medicinal Plants Studies, Yaounde, Cameroon, **5** Department of Microbiology, The University of Yaounde I, Yaounde, Cameroon, **6** Camdiagnostic, Ministry of Scientific Research and Innovation, Yaounde, Cameroon, **7** Bacteriology Department, Centre Pasteur of Cameroon, Yaounde, Cameroon, **8** Centre de Recherche sur les Maladies Émergentes et Re-Emergentes, Institut de Recherches Médicales et d'Etudes des Plantes Médicinales, Yaounde, Cameroon, **9** Department of Animals Biology and Physiology, The University of Yaounde I, Yaounde, Cameroon, **10** Department of Biochemistry, The University of Yaounde I, Yaounde, Cameroon, **11** Virology Laboratory, Chantal Biya International Reference Center for Research on HIV/AIDS Prevention and Management, Yaounde, Cameroon, **12** Research Monitoring and Planning Unit, National Aids Control Committee, Douala, Cameroon, **13** Department of Environment and Health, Istituto Superiore di Sanità, Rome, Italy

* kenmoe@pasteur-yaounde.org (SK); njouom@pasteur-yaounde.org (RN)

## Abstract

A substantial amount of epidemiological data has been reported on Enterovirus D68 (EV-D68) infections after the 2014 outbreak. Our goal was to map the case fatality rate (CFR) and prevalence of current and past EV-D68 infections. We conducted a systematic review (PROSPERO, CRD42021229255) with published articles on EV-68 infections in PubMed, Embase, Web of Science and Global Index Medicus up to January 2021. We determined prevalences using a model random effect. Of the 4,329 articles retrieved from the data-bases, 89 studies that met the inclusion criteria were from 39 different countries with apparently healthy individuals and patients with acute respiratory infections, acute flaccid myelitis and asthma-related diseases. The CFR estimate revealed occasional deaths (7/1353) related to EV-D68 infections in patients with severe acute respiratory infections. Analyses showed that the combined prevalence of current and past EV-D68 infections was 4% (95% CI = 3.1–5.0) and 66.3% (95% CI = 40.0–88.2), respectively. The highest prevalences were in hospital outbreaks, developed countries, children under 5, after 2014, and in patients with acute flaccid myelitis and asthma-related diseases. The present study shows sporadic



**Data Availability Statement:** All relevant data are within the manuscript and its Supporting information files.

**Funding:** This project is part of the EDCTP2 programme supported by the European Union under grant agreement TMA2019PF-2705 to SK (http://www.edctp.org/projects-2/edctp2-projects/edctp-preparatory-fellowships-2019/#). The funders had no role in study design, data collection and analysis, decision to publish, or preparation of the manuscript.

**Competing interests:** The authors have declared that no competing interests exist.

deaths linked to severe respiratory EV-D68 infections. The study also highlights a low prevalence of current EV-D68 infections as opposed to the existence of EV-D68 antibodies in almost all participants of the included studies. These findings therefore highlight the need to implement and/or strengthen continuous surveillance of EV-D68 infections in hospitals and in the community for the anticipation of the response to future epidemics.

## Author summary

Enterovirus D68 (EV-D68) infections represent a global public health concern. EV-D68 are detected in apparently healthy subjects and patients with acute respiratory illnesses, acute flaccid myelitis, and asthma-related illnesses. Enterovirus D68 was first described in 1962 and exhibited sporadic circulation until August 2014 when outbreaks of EV-D68 infections were reported in the USA and Canada mainly in children with acute flaccid myelitis and severe acute respiratory disease. We systematically reviewed the literature on EV-D68 infections globally in the present study to determine the case fatality rate and prevalence of current and past infections. Our results show sporadic deaths in patients with severe acute respiratory EV-D68 infections. Our data also show a low prevalence of EV-D68 in current infections unlike the presence of EV-D68 antibodies (past infections) in almost all individuals of all ages. EV-D68 infections were more prevalent in hospital outbreaks, industrialized countries, children < 5 years, and patients with acute flaccid myelitis and asthma-related diseases. These data highlight the need to strengthen the surveillance of EV-D68 infections.

## Introduction

Enteroviruses (EVs) are a major public health concern worldwide. Although the majority of EV infections are subclinical, they can be associated with a broad spectrum of clinical illnesses mainly in children, including acute respiratory illness, meningitis, encephalitis, myocarditis, pericarditis, conjunctivitis, gastrointestinal diseases, hand-foot-and-mouth disease (HFMD), inflammatory muscle diseases, and rarely, its can lead to neurological complications in severe cases [1–3]. Enteroviruses are members of the genus *Enterovirus* (*Picornaviridae* family). They are non-enveloped, positive-sense single strand RNA viruses with of approximately 7400–7500 nucleotides in length. More than 100 Enterovirus genotypes are currently classified based on their molecular and antigenic properties into seven types associated with human disease (Enterovirus A-D and Rhinovirus A-C) [3,4]. While Human rhinoviruses tropism is restricted to the respiratory tract, the vast majority of Human enteroviruses infect the gastrointestinal tract, the central nervous system, respiratory tract and other organs such as heart and cause a significant morbidity and mortality worldwide.

First isolated in California, in 1962 [5], enterovirus D68 (EV-D68) which belongs to the species enterovirus D, circulated at low levels around the world all over the year and infections had been identified sporadically until 2014 [6–8] and only 699 persons were confirmed infected worldwide before 2014 [7]. In August 2014, virus has emerged and captured public attention when a widespread outbreak of neurological impairment (mostly acute flaccid myelitis (AFM)) and severe respiratory illness has firstly been reported across United States and Canada and affecting more than 2000 persons worldwide, mainly children [5,7,9–18]. These severe illness requiring hospitalization and admissions to intensive-care units were potentially

fatal particularly in children [7,16,18–20]. This unpredictable outbreak of enterovirus D68 associated to severe respiratory illness and AFM in 2014, led to implementation of enhanced laboratory-based surveillance for enterovirus in many countries.

Following the 2014 outbreak, other waves of EV-D68 infections were observed in 2016 and 2018 with reports of outbreaks in several parts of the world, including Europe [10,21–26], USA [15,27,28], South America [29,30], Asia [31,32] and West Africa [33,34]. These EV-D68 outbreaks also coincided with waves of AFM. Thus, AFM mainly associated with EV-D68 is now recognized as a global disease, with hundreds of cases reported across Europe [35], Asia [36,37], Australia [38], Africa [33,34], North America [39] and South America [30,40].

Given that there are no vaccines approved to protect against EV-D68 infections and given its ability to cause a potential uncontrollable epidemic of severe respiratory illness and potential neurological complications such as AFM, it is important to understand its full epidemiology spectrum, in order to elucidate the patterns and outbreak dynamics of such infections and predict its long-term disease burden and impact on public health. In this review, we seek to systematically review the available literature and assess EV-D68 infections prevalence and case fatality rate in humans.

## Methods

### Protocol and registration

The Checklist of Preferred Reporting Items for Systematic Reviews and Meta-Analysis (PRISMA) was used in the development and reporting of this review [41] (S1 Table). The PROSPERO international prospective registry was used for the registration of the protocol (ID: CRD42021229255).

### Data sources and search strategies

Extensive searches were performed by an investigator in the PubMed, Embase, Web of Science and Global Index Medicus databases from inception to January 05, 2021. To enhance the sensitivity of the search strategy, we used only terms covering the field of EV-D68 infections (S2 Table). The search strategy was initially applied to PubMed and then adapted to other databases. Other potentially eligible studies were searched from the list of references of included studies.

### Study eligibility

Titles and abstracts were independently reviewed in Rayyan review tool by two investigators to rule out those who did not meet the inclusion criteria. Studies that reported data in English and/or French on the case fatality rate (CFR) and/or prevalence of current or past EV-D68 infections were included. No restrictions on study design, type of EV-D68 detection method, type of specimen for testing for EV-D68, geographic regions or population category were applied. Studies with selection bias (selection of samples with preliminary Rhinovirus (RV) and/or EV results known for EV-D68 typing), with full texts and/or abstracts not found, duplicates, and with sample size <10 were excluded. Studies in which all generic RT-PCR positives for EV/RV were not typed for EV-D68 were excluded for inappropriate prevalence.

### Data extraction

Study eligibility and data retrieval was performed in duplicate from full text articles using Google form. Disagreements were resolved by discussion and consensus among independent investigators. Pre-designed and pre-tested google forms were used to collect the following

data: first author name, year of publication, study design, sampling method, number of sites, time of sample collection, country, study period, age group, population studied, EV-D68 diagnostic method, sample types, number tested for EV-D68, number with EV-D68, number of deaths among EV-D68 positive, and data from the assessment of risk of bias according to Hoy et al (S3 Table) [42]. We collected only the first EV-D68 specific screening technique and did not consider the confirmatory screening methods such as sequencing. In studies reporting the EV-D68 prevalence of several different sample types or detection methods for the same participants, we selected the highest prevalence. A single study could contribute to multiple points of prevalence specific to country, population category, sample type and detection technique.

## Data synthesis and analysis

We categorized the countries according to the United Nations Statistics Division (UNSD), World Health Organization (WHO) and income level [43–45]. The age range was grouped into <5 years, <18 years, and all ages. Population categories included presumably healthy individuals, patients with acute respiratory infections (ARI), acute flaccid myelitis (AFM), and asthma-related illnesses. Patients with respiratory symptoms were considered to be ARI. Patients with respiratory symptoms accompanied by hospitalization or lower respiratory tract infections were considered as severe acute respiratory infections (SARI). Infections were aggregated into current (detection of live virus, RNA and/or viral antigen) and past (detection of antibodies). We presented the individual data of the included studies in a database. We aggregated these individual data with the qualitative variables in number and percentage and the continuous variables in range. Data analyses were carried out using R software version 4.0.3 [46,47]. The forest plot, the proportions grouped with the 95% confidence interval were obtained by a random-effect meta-analysis [48,49]. Estimates were obtained following a Freeman-Tukey double arcsine transformation. Cochran's Q test and the I2 statistic were used to assess the possibility of heterogeneity between studies, with the values of p <0.05 and I2> 50% indicating the presence of heterogeneity [48,50]. The funnel plot and Egger's test were used to assess publication bias, with the skewness of the funnel diagram and the p-value <0.05 indicating the presence of publication bias [51]. The stability of the meta-analyses was assessed by sensitivity analyses including only cross-sectional studies and studies with a low risk of publication bias. Subgroup analyses were conducted according to study design, sampling, timing of sample collection, country, WHO region, UNSD region, country income level, period of study (before and after 2014), age group, category of study population, EV-D68 diagnostic method, and sample type. Subgroup analyses were only conducted when at least 3 studies that fell into at least two different categories applied.

## Results

### Study selection and characteristics

The article selection process is shown in Fig 1. The database search returned 4329 articles including 992 duplicates. A total of 89 articles were included in this meta-analysis after exclusions by title and abstract and full-text eligibility review (S4 Table) [11,16–18,20,25,30,32–34,37,52–129]. These 89 articles were published between 2011 and 2020 in 39 countries in the 6 WHO regions and mainly in high-income countries (USA, China and Japan) (S5 and S6 Tables). Participants were recruited between 1994 and 2019, but most of the participants were after 2014, when EV-D68 infections emerged in the USA. Most of the studies were cross-sectional and prospective with non-probabilistic recruitment. The recruited participants consisted of presumed healthy individuals and patients with acute respiratory infections, acute flaccid myelitis and asthma-related illnesses. The predominant technique used was RT-PCR to

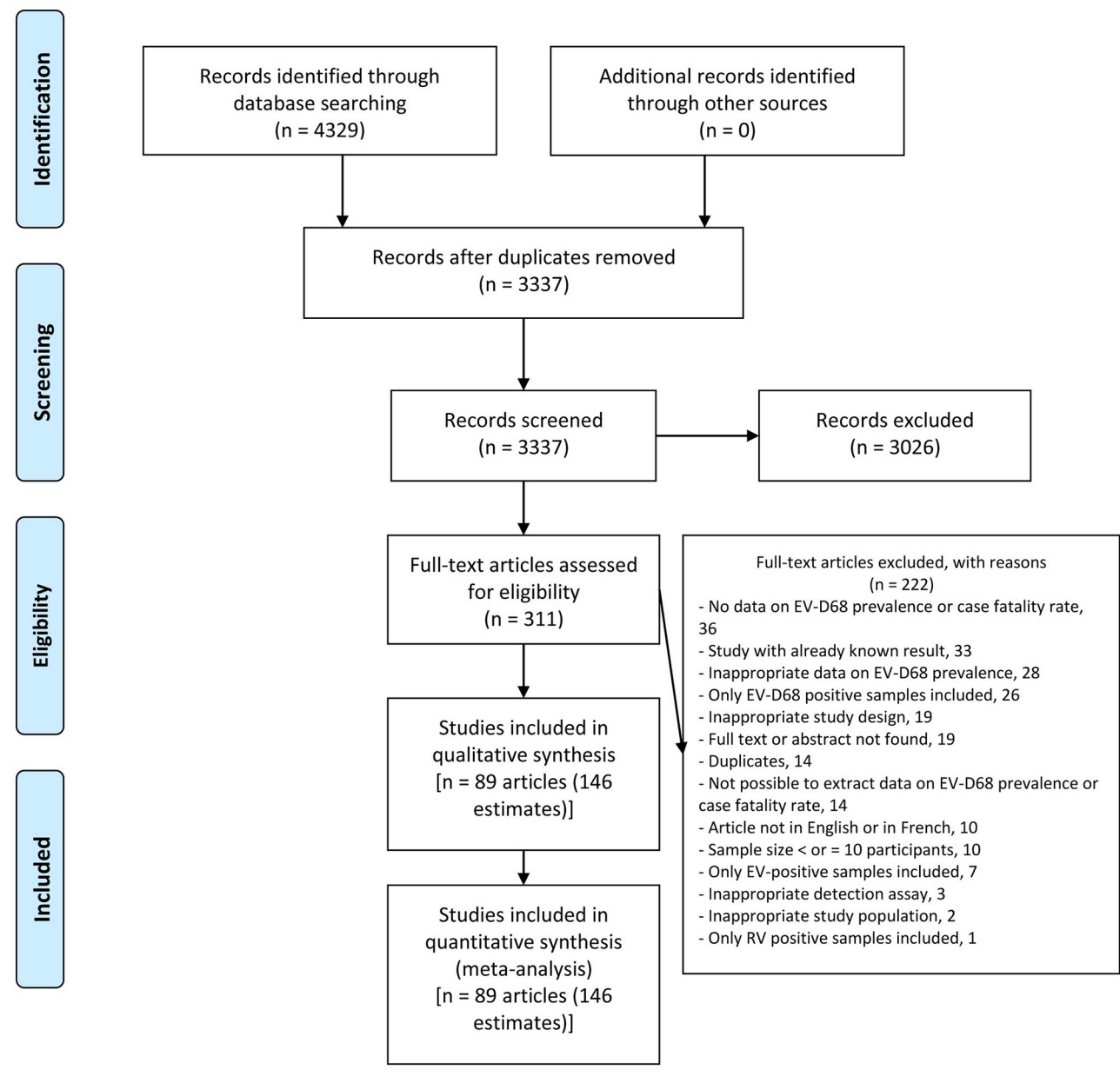

**Fig 1. Flow diagram summarizing the selection of eligible studies.**

search for EV-D68 RNA in nasopharyngeal samples. About half of the studies had a low risk of bias and no study had a high risk of bias (S7 Table).

## The case fatality rate of Enterovirus D68 (Meta-analysis)

The study with reported death due to EV-D68 infections were conducted in 5 countries: Canada, Japan, Panama, United Kingdom and USA (Fig 2). Seven deaths in patients with ARI were reported among 1353 participants in two included studies conducted in the United Kingdom and USA [20,58]. The CFR due to EV-D68 infections ranged from 0 to 4.4% in the 10 included studies with a combined random effect rate of 0.0% (95% CI: 0.0–0.1) (Fig 3)

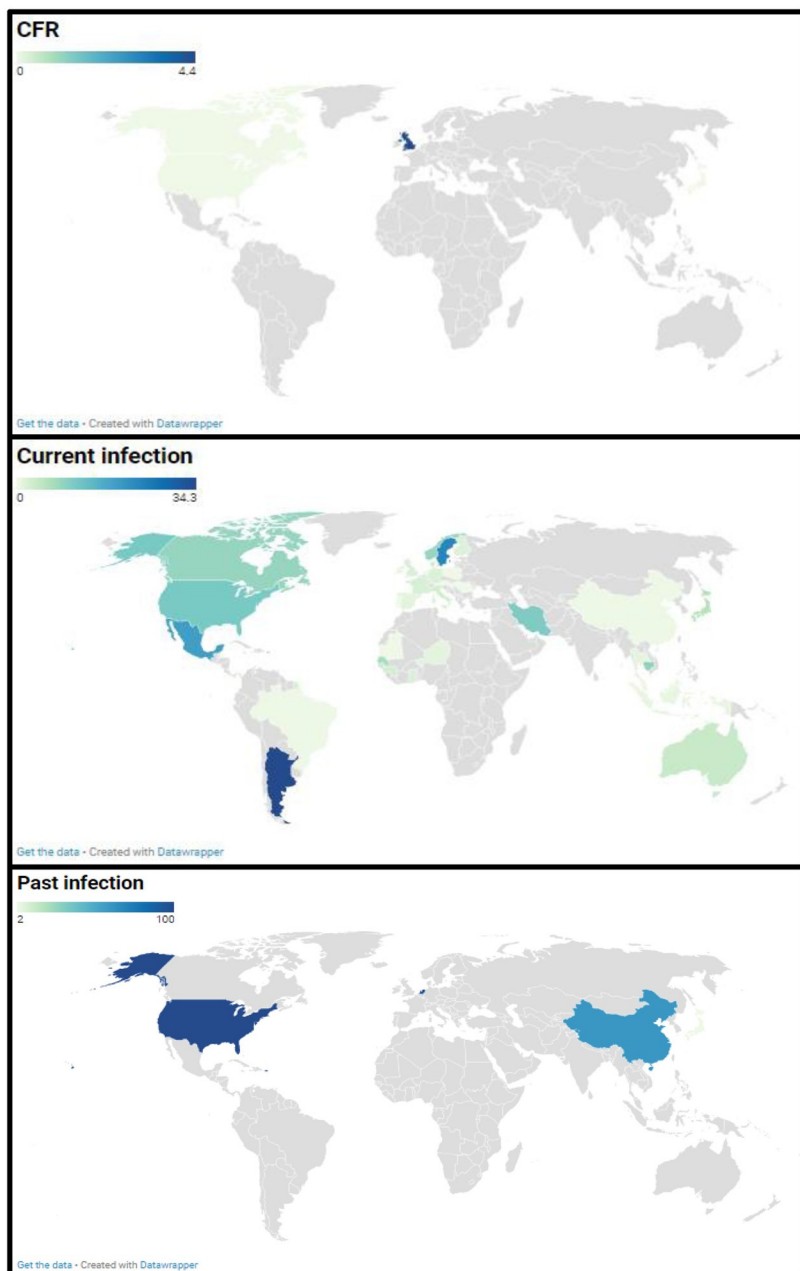

**Fig 2. Global distribution, case fatality rate, and prevalence estimate of Enterovirus D68 current and past infections.** The figures represent from top to bottom: the Enterovirus D68 case fatality rate and the prevalence of Enterovirus D68 current and past infections. Map source: https://www.datawrapper.de/basemaps/world/.

[18,20,53,58,59,65,102,112,113,127]. No statistical heterogeneity was observed in the overall estimate of the CFR (I2 = 27.5% [0.0%; 65.2%]).

### The prevalence of Enterovirus D68 current infection (Meta-analysis)

Studies reporting EV-D68 current infections were conducted in 41 countries across 5 WHO regions: America, Europe, West Pacific, Africa, and Southeast Asia (Fig 2). A total of 4,440 EV-D68 infections were reported among 204,351 participants with acute respiratory

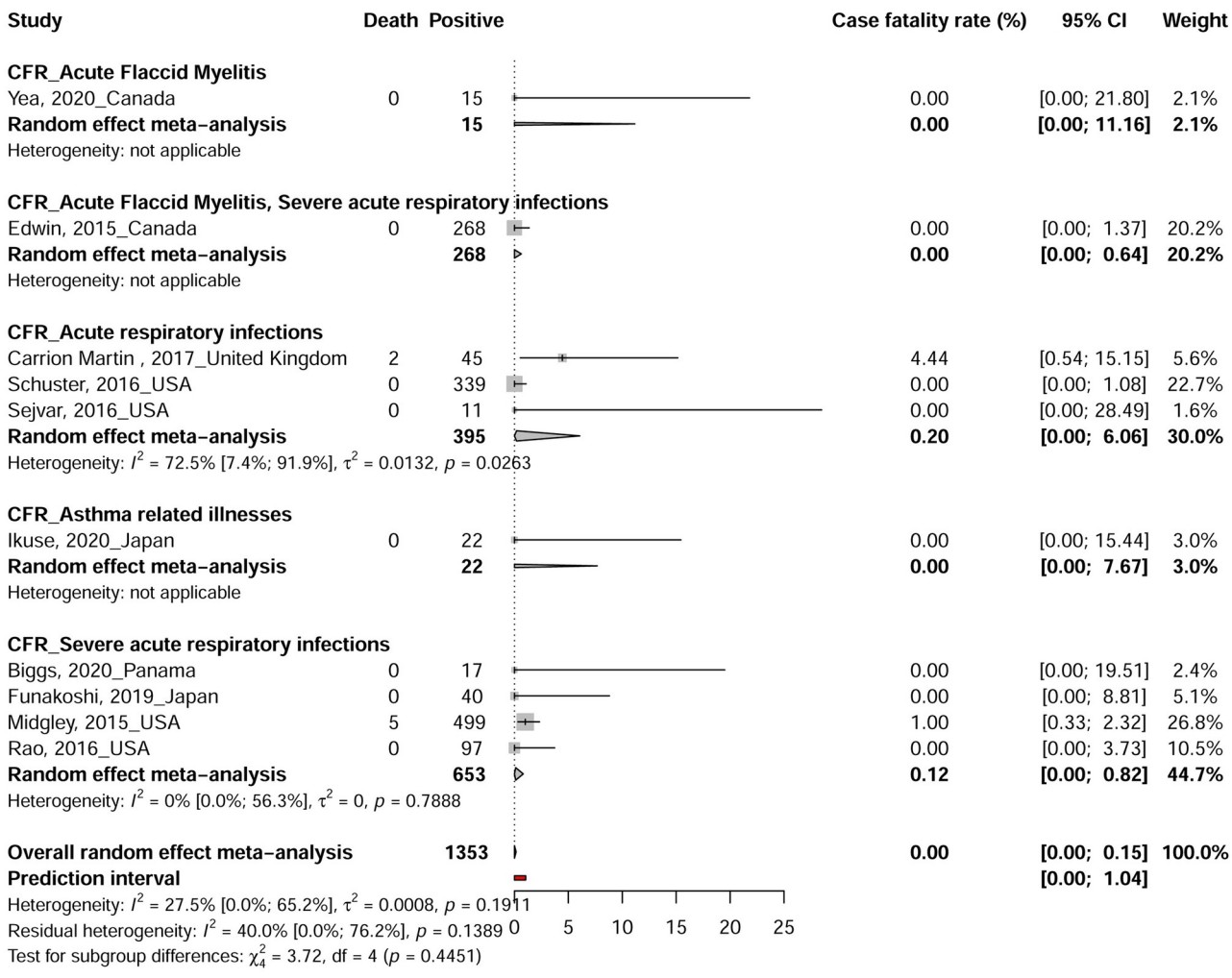

**Fig 3. The pooled global case fatality rate of Enterovirus D68.**

infections, acute flaccid myelitis, asthma-related illnesses, and presumed healthy individuals. EV-D68 current infection prevalences ranged from 0 to 74.3% in the included studies with a combined random effect rate of 4.0% (95% CI: 3.1–5.0) (Fig 4). Substantial statistical heterogeneity was observed in the overall estimate of the prevalence of EV-D68 current infections ($I2$ = 99.0% [98.9%; 99.1%]).

## The prevalence of Enterovirus D68 past infection (Meta-analysis)

Studies reporting EV-D68 past infections were conducted in 4 countries China (four estimates), USA (one estimate), Japan (one estimate) and the Netherlands (one estimate) (Fig 2). A total of 2390 EV-D68 past infections were reported among 3004 participants with acute flaccid myelitis and presumed healthy individuals in 7 included studies [37,60,63,69,76,79,115]. EV-D68 past infections prevalences ranged from 2.0 to 100.0% in included studies with a combined random effect rate of 66.3% (95% CI: 40.0–88.2) (Fig 5). Substantial statistical heterogeneity was observed in the overall estimate of the prevalence of EV-D68 past infections ($I 2$ = 99.5% [99.3%; 99.6%]).

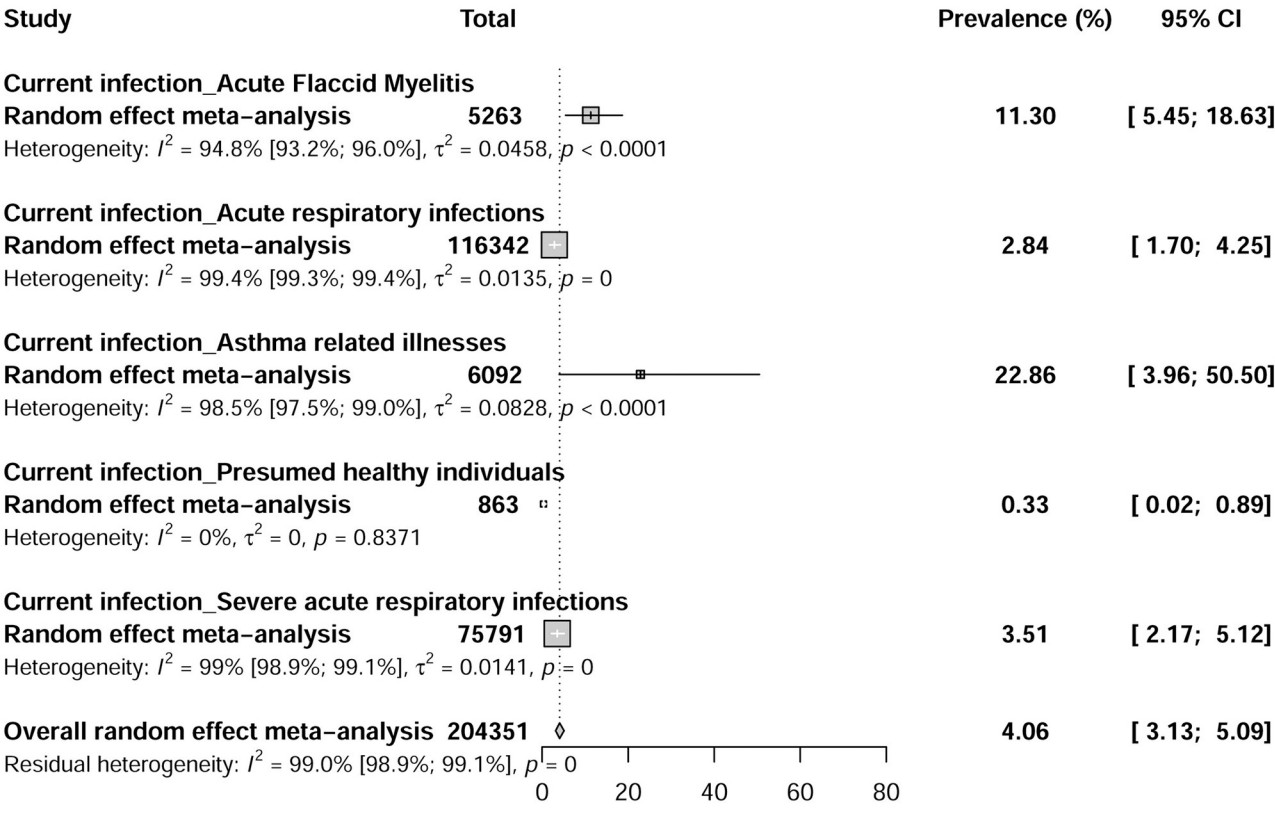

**Fig 4. The pooled global prevalence of Enterovirus D68 current infections.**

## Publication bias and sensitivity analysis

Egger's test indicates that no publication bias was observed for the estimates of CFR and prevalence of EV-D68 past infections (Table 1). Egger's test indicates publication bias in estimating

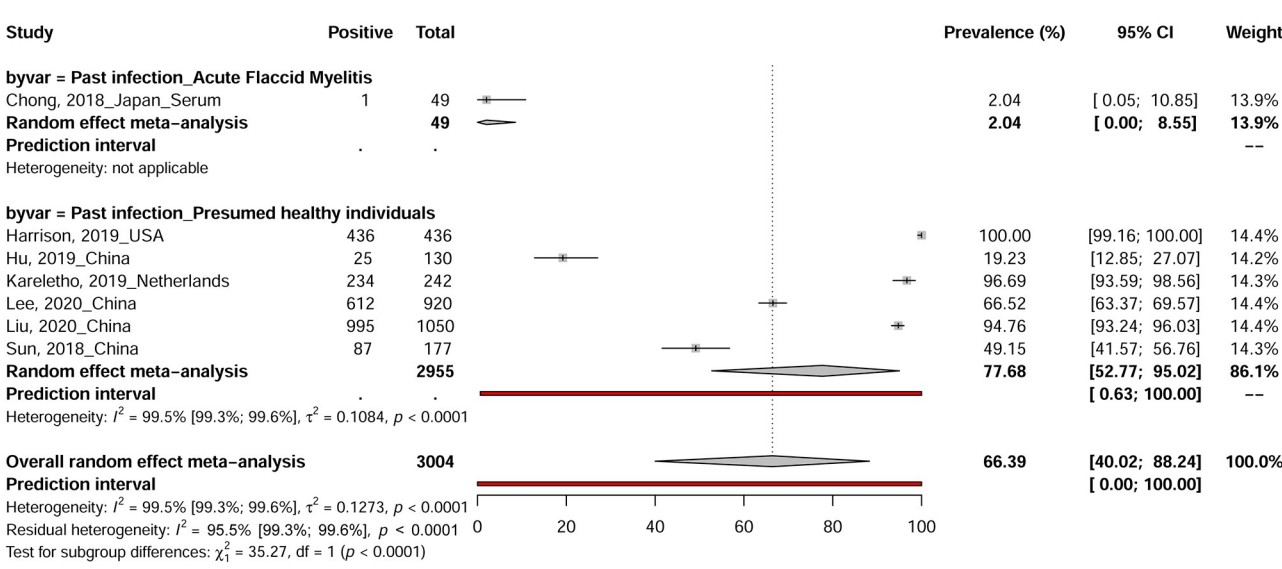

**Fig 5. The pooled global prevalence of Enterovirus D68 past infections.**

**Table 1. Summary of meta-analysis results for global case fatality rate and prevalence of Enterovirus D68 in humans.**

| | Prevalence. % (95%CI) | 95% Prediction interval | N Studies | N Participants | ¶H (95%CI) | §I² (95%CI) | P heterogeneity | P Egger test |
|---|---|---|---|---|---|---|---|---|
| | | | EV-D68 case fatality rate | | | | | |
| Overall | 0 [0–0.2] | [0–1] | 10 | 1353 | 1.2 [1–1.7] | 27.5 [0–65.2] | 0.191 | 0.343 |
| Cross-sectional | 0 [0–0.1] | [0–0.5] | 7 | 1271 | 1.1 [1–1.5] | 14.7 [0–58.4] | 0.318 | 0.862 |
| Low risk of bias | 0 [0–0.7] | [0–1.4] | 5 | 587 | 1 [1–1.8] | 0 [0–70.6] | 0.586 | 0.345 |
| | | | EV-D68 prevalence | | | | | |
| **Current infection** | | | | | | | | |
| Overall | 4.1 [3.1–5.1] | [0–18.1] | 107 | 204351 | 9.9 [9.6–10.3] | 99 [98.9–99.1] | < 0.001 | < 0.001 |
| Cross-sectional | 3.8 [2.9–4.8] | [0–17.6] | 102 | 199378 | 10.1 [9.7–10.4] | 99 [98.9–99.1] | < 0.001 | < 0.001 |
| Low risk of bias | 2.4 [1.7–3.3] | [0–11.2] | 55 | 165151 | 9.6 [9.1–10.1] | 98.9 [98.8–99] | < 0.001 | 0.002 |
| **Past infection** | | | | | | | | |
| Overall | 77.7 [52.8–95] | [0.6–100] | 6 | 2955 | 13.8 [12.2–15.7] | 99.5 [99.3–99.6] | < 0.001 | 0.521 |
| Cross-sectional | 93.2 [74.5–100] | [0–100] | 4 | 2648 | 13.3 [11.2–15.7] | 99.4 [99.2–99.6] | < 0.001 | 0.540 |
| Low risk of bias | 77.7 [52.8–95] | [0.6–100] | 6 | 2955 | 13.8 [12.2–15.7] | 99.5 [99.3–99.6] | < 0.001 | 0.521 |

CI: confidence interval; N: Number; 95% CI: 95% Confidence Interval; NA: not applicable.

¶H is a measure of the extent of heterogeneity, a value of H = 1 indicates homogeneity of effects and a value of H >1indicates a potential heterogeneity of effects.

§: I2 describes the proportion of total variation in study estimates that is due to heterogeneity, a value > 50% indicates presence of heterogeneity

the prevalence of EV-D68 current infections. The distribution of the studies in the funnel plot was in agreement with the results of the Egger test (S2–S4 Figs). The results of sensitivity analyses including only cross-sectional studies and studies with low risk of bias showed no significant difference from the overall estimates.

## Sub-group analysis

Relationships between EV-D68 CFR and prevalence (current and past infections) according to study design, sampling approach, timing of sample collection, country, WHO region, UNSD region, country income level, study period, age group, category of study population, diagnostic method, and sample type were established with subgroup analyses (S8 Table). The CFR was not heterogeneous and none of these factors significantly affected the reported estimates. The prevalence of EV-D68 current infections were significantly higher in hospital outbreak, cohorts, study with non-probabilistic recruitments, prospective studies, countries in the Americas WHO region (USA, Argentina, Mexico, and Canada), high income countries, studies with recruitments made after August 2014, studies in <5 years old, studies recruiting patients with asthma-related diseases and AFM and studies that detected EV-D68 in nasopharyngeal samples by real-time RT-PCR. The prevalence of EV-D68 past infections were significantly higher in studies with non-probabilistic recruitments, in the USA, in high-income countries and in subjects of all ages.

## Discussion

This systematic review provides a summary of the prevalence of EV-D68 past and current infections and its case fatality rate in humans obtained by the exploitation of published articles between 1962 and 2020 in 41 countries across 5 WHO regions: America, Europe, West Pacific, Africa, and Southeast Asia. This review reveals occasional deaths from patients with acute

respiratory EV-D68 infections. The combined prevalence of current EV-D68 infections in apparently healthy individuals and patients with ARI, AFM and asthma-related illness was 4%. Current EV-D68 infections were higher in hospital outbreaks, high-income countries in the Americas, in children <5 years, after 2014, and in patients with AFM and asthma-related illnesses. Almost all of the individuals had scars from EV-D68 past infections with a combined prevalence of 66.3%. EV-D68 past infections were present in subjects of all ages and higher in high incomes countries.

Although, there is no approved antiviral for EV-D68 infections, this review shows a low case fatality rate with seven deaths in patients with ARI reported among 1353 participants. This low number of deaths can be explained by early management with supportive treatment with careful monitoring focused on potential vital emerging complications. In addition, EV-D68 infections are mainly reported in high-income countries, which may result a better care. In 2018, 96% of identified AFM cases in the USA were admitted to hospital, and 58% to an intensive-care unit [130]. Nonetheless, Enterovirus D68 treatment option is limited to supportive care for mild and severe cases. Sufficient evidence of efficacy of corticosteroids, immunoglobulins or plasma exchange has not been found although these immunomodulatory treatments have often administered during the acute management of this pathology [113]. Fluoxetine was found to have in vitro activity against EV-D68 infections. A recent study on the use of fluoxetine in patients with AFM found that fluoxetine was well tolerated and safe. In addition, immune suppressive therapies have also been used by some centers to target the inflammatory response to infection [131]. It should be noted that recently much progress has been made in the development of the EV-D68 antivirals by targeting various viral proteins and host factors essential for viral replication [132]. However, this intervention was not associated with improved neurologic outcomes [133].

Overall EV-D68 current infection prevalence around 4.0% was estimated in this study with a range of prevalence of 0.0 and 74.29%. The wide range of prevalence found can be explained by the large variation of the prevalence of EV-D68 infections observed according to the study period, the epidemic context, the geographical location, the target population, as well as the various detection assays. High infection rate in North America may be explained by the fact that outbreaks occurred there before spreading in other geographies. The high prevalence of EV-D68 (current and past infection) observed in developed countries could also be attributed to the establishment of an effective surveillance system and the numerous studies that are carried out. In Europe, increasing awareness of and testing for enterovirus D68 led to 31 enterovirus D68-associated AFM cases reported in 2016, an increase of greater than 10 times compared with 2014 [134]. In contrast, in developing countries such as African countries, the low prevalence of EV-D68 infections can be explained by the unawareness of the virus and the resulting lack of studies carried out. Thus, there may be a silent circulation of the virus as witnessed by retrospective studies carried out in West Africa in 2019 [33,135,136] and 2020 [34] which showed circulation of the virus during the epidemics of 2014 and 2016.

In this review the majority of EV-D68 current infections were recorded after the 2014 epidemic. Indeed, minor outbreaks of enterovirus D68 had also been described, with 699 cases confirmed in Europe, Africa, America and Asia from 1970 to 2013 with disease manifestations mainly ranging from mild respiratory symptoms to severe ARI requiring intensive care [7]. The high numbers of EV-D68 infections reported after 2014 in the review can therefore be explain by the multiple outbreaks observed from 2014. Indeed, only during the 2014 EV-D68 infections outbreak, 2287 cases recorded in 20 countries which are mainly located in North America and Europe, but also in Southeast Asia and Chile [7]. Following the 2014 outbreak, an upsurge of EV-D68 infections was also occurred in 2016 and 2018 in almost geographic location [10,15,22,24,28,32–34,137]. Regarding the study period, since 2014, previously

published data on the seasonality of EV-D68 infections have suggested circulation with a biennial epidemic cycle of EV-D68 infections in Europe and North America [70,138,139]. Moreover, the periodic upsurge of EV-D68 infections in same geographical area is supporting by study of Gilrane and colleagues which reported that the resurgence of EV-D68 infections was only recognized in 2014, 2016 and 2018 compared to 2015 and 2017 where only sporadic cases were detected [140]. The impact of geographical location in prevalence can be explained in USA context were several studies noticed a widespread of EV-D68 infections in 2014 and 2018 but only low-level circulation in 2016 in some states, such as Colorado, Missouri, and Ohio. In contrast, high prevalence of EV-D68 infections was reported in the Lower Hudson Valley and Philadelphia in 2016 [139,140].

The high prevalence observed in hospital outbreak in this study can be explained by nosocomial transmission of the virus. Indeed, evidence supporting the role of EV-D68 in respiratory infections in hospitalised patients has been provided in several countries as Spain [141], in Italy [24], Japan [142] and USA [143].

As in our study, it has been reported in several studies that the paediatric population is more vulnerable to EV-D68 infections [22,33,70]. EV-D68 infections has mostly been reported in children probably because of the lower amount of physical space in the airways than in adults but also the immaturity of their immune system [19,78,87,106,144]. Although children are at higher risk for severe respiratory symptoms than are adults [64,78,144–148], cases of respiratory disease associated with EV-D68 have been reported in both healthy adults and adults with under lying respiratory diseases or immunosuppression [149]. However, although constituted of four clades and five subclades, including clade A, clade B (subclades B1 to B3), clade C and clade D (subclades D1 to D2), recent studies reported that EV-D68 clade D1 is more likely to cause respiratory infections in adults than in children [26,96,115,150]. Nonetheless, AFM occurs almost exclusively in children [151]. Indeed, these major co-circulating EV-D68 (AC) clades emerged in the world in the 2000s [6] and subsequently diversified, with a single monophyletic group (genotypes B1 and B3) with a common ancestor in 2009 associated until now at AFM [152]. In vitro studies of the neurotropism of these viruses compared to ancestral strains have given conflicting results as to whether neurotropism increased [153,154]. The timing of increased transmission estimated in a recent study in the UK on the basis of serological data analysis roughly corresponds to the genetic emergence of clade B around 2007, and it could be hypothesized that increased transmissibility of the virus is a trait associated with this clade [155]. More efficient viral replication may improve transmission as well as the likelihood of the virus reaching the central nervous system, although changes in receptor use may also play a role [155]. Regarding immunity, studies that more closely investigated EV-D68 infections seroepidemiology in children noted a low population antibody prevalence around one year of age, with a gradual rise in the presence of EV-D68 neutralizing antibodies through childhood [115,156]. A recent study shown also that natural EV-D68 infections of humans induces B cells encoding broad and potently neutralizing antibodies that can prevent or treat infection and disease in both the respiratory tract and the nervous system [157]. In addition, EV-D68 phylogenetic of pattern does not suggest continuous immune escape [158] and a broad range of cross-reactivity among clades of EVD68, both with binding and neutralization was also observed by Vogt et al. [157].

High prevalence of EV-D68 infections in people with asthma was observed in this study. Indeed, it has been shown that EV-D68 infections peaks in 2014 and 2018 were related to an increase in summertime hospitalizations for asthma in USA, which agrees with a study conducted in Japan [145]. Some studies have shown a more severe clinical course among hospitalized children with EV-D68 infections who have pre-existing asthma [20,159], whereas others have not observed this difference [16,160].

In this study, from 5263 participants with AFM, an overall prevalence of EV-D68 current infections of 11.3% was obtained. However, although other pathogens like enterovirus A71 and West Nile virus have been recognized responsible to AFM, EV-D68 is associated for the majority of AFM cases. However, several evidences have been shown which support that the AFM was caused by EV-68. The first is the coincidence of outbreaks of EV-68 and the increase of AFM case in 2014, 2016 and 2018 [35,37,156]. In addition, EV-D68 was found predominantly in specimens from patients with AFM [52,113,151]. The second evidence has been highlighted by two independent studies on sera and CSF of AFM patients who have demonstrated a strong antibody enrichment associated with enteroviruses, including antibodies against a specific epitope of the EV-D68 in up to 73% of patients [161,162]. In addition, multiple mouse models of EV-D68 have been developed that replicate key features of AFM [163–167] and Koch's postulates for causation were fulfilled in one mouse model [164]. Two others analyses have also concluded that the Bradford Hill criteria support a causative relationship between EV-D68 and AFM in humans [39,168]. Moreover, these facts may explain the high prevalence of EV-D68 in AFM, which corroborates with the high prevalence of 22.5 and 27.8 observed in California from 2012 to 2015 [118] and in Canada in 2014 and 2018 [127], respectively.

Past infections were investigated in this review in patients with acute flaccid myelitis and presumed healthy individuals and EV-D68 past infections prevalence of 66.3% was found with a range from 2.0 to 100.0%. As the result, anti-EV-D68 neutralizing antibodies levels have been gradually increasing worldwide with a geometric mean titer 40.50 in 2002 in the Finnish [1] and a highest antibody titer reached 676.08 in the Kansas City population from 2011 to 2013 [115]. In additions, after the 2014 EV-D68 infections outbreak in the United States, the geometric means of the titers of anti-EV-D68 neutralizing antibodies in human sera was up to 14, 562 [169]. Moreover, the presence of neutralizing antibodies is a widely accepted correlate of immunity and protection against severe disease associated with EV infection [170]. Thus, age-stratified serosurveys of neutralizing antibodies are a valuable method of understanding the prevalence of EV-D68 infections and evaluating the risk of an outbreak among the general population.

One of the main limitations of this study is the poverty or lack of data observed in resource limited settings. Furthermore, the role played by EV-D68 in the deaths reported in this review is unclear. Most of the published work included in this review retrospectively tested EV-D68 in nasopharyngeal swabs and therefore could likely underestimate the reported prevalences. The heterogeneity between diagnostic tools used in different included studies also constitutes a limit in the detection of EV-D68. Regarding the diagnostic, in this study we also considered all AFP as AFM. However, a consensus case definition applicable in all WHO regions will help differentiate between AFM and other neurological complications. Although there were some limitations, this is to the very best of our knowledge, the first review with meta-analysis summarizing data on the global prevalence of EV-D68 infections. This review also allowed us to map the articles published on the EV-D68 infections and to see the data gap observed in countries with limited resources. This article also highlighted the importance of AFM cases associated with EV-D68, nosocomial infections and the vulnerability of children with asthma.

In the absence of a vaccine, prevention is based on strict respect with universal hygiene rules, and in particular hand washing, to stop the chain of transmission within the family and the community. EV-D68 infections outbreak observed in hospital in this study demands effective implementation of vigorous prevention and control strategies to break nosocomial EV-D68 infections. For that personal protective equipment (PPE), like face masks, will help to prevent the spread EV-D68 infections. In addition, until more progress in treatment is

achieved; we recommend that the children population should remain shielded during EV-D68 infection outbreaks. For a better understanding of the virus and the diseases caused, efficient and standardized laboratory diagnosis and characterization of circulating viral strains is the first step towards effective and continuous surveillance activities.

In conclusion, the high prevalence observed in these studies shows that EV-D68 should be considered a pathogen emerging. The large-scale outbreak of 2014, 2016 and 2018, sometimes associated with respiratory infections complicated by severe neurological damage, must encourage the implementation of continuous global surveillance. However, standardization of detection systems, types of samples and case definitions, especially for AFM, would be a real advance in understanding about the mechanisms for the sudden upsurge in incidence and its unusual and severe complex disease manifestations.

## Supporting information

**S1 Table. Preferred reporting items for systematic reviews and meta-analyses checklist.**
(PDF)

**S2 Table. Search strategy in PubMed.**
(PDF)

**S3 Table. Items for risk of bias assessment.**
(PDF)

**S4 Table. Main reasons of exclusion of eligible studies.**
(PDF)

**S5 Table. Characteristics of included studies.**
(PDF)

**S6 Table. Individual characteristics of included studies.**
(PDF)

**S7 Table. Risk of bias assessment.**
(PDF)

**S8 Table. Subgroup analyses of worldwide case fatality rate and prevalence of Mycobacterium ulcerans in humans, animals, plants, and environment.**
(PDF)

**S1 Fig. The pooled global prevalence of Enterovirus D68 current infections.**
(PDF)

**S2 Fig. Funnel chart for publications of the Enterovirus D68 case fatality rate.**
(PDF)

**S3 Fig. Funnel chart for publications of the prevalence of Enterovirus D68 current infections.**
(PDF)

**S4 Fig. Funnel chart for publications of the prevalence of Enterovirus D68 past infections.**
(PDF)

## Acknowledgments

None.

## Author Contributions

**Conceptualization:** Amary Fall, Sebastien Kenmoe, Richard Njouom.

**Data curation:** Amary Fall, Sebastien Kenmoe, Jean Thierry Ebogo-Belobo, Donatien Serge Mbaga, Arnol Bowo-Ngandji, Joseph Rodrigue Foe-Essomba, Serges Tchatchouang, Marie Amougou Atsama, Jacqueline Félicité Yénigué, Raoul Kenfack-Momo, Alfloditte Flore Feudjio, Alex Durand Nka, Chris Andre Mbongue Mikangue, Jean Bosco Taya-Fokou, Jeannette Nina Magoudjou-Pekam, Efietngab Atembeh Noura, Cromwel Zemnou-Tepap, Dowbiss Meta-Djomsi, Martin Maïdadi-Foudi, Ginette Irma Kame-Ngasse, Inès Nyebe, Larissa Gertrude Djukouo, Landry Kengne Gounmadje, Dimitri Tchami Ngongang, Martin Gael Oyono, Cynthia Paola Demeni Emoh, Hervé Raoul Tazokong, Gadji Mahamat, Cyprien Kengne-Ndé, Serge Alain Sadeuh-Mba.

**Formal analysis:** Sebastien Kenmoe.

**Methodology:** Amary Fall, Sebastien Kenmoe, Jean Thierry Ebogo-Belobo, Donatien Serge Mbaga, Arnol Bowo-Ngandji, Joseph Rodrigue Foe-Essomba, Serges Tchatchouang, Marie Amougou Atsama, Jacqueline Félicité Yénigué, Raoul Kenfack-Momo, Alfloditte Flore Feudjio, Alex Durand Nka, Chris Andre Mbongue Mikangue, Jean Bosco Taya-Fokou, Jeannette Nina Magoudjou-Pekam, Efietngab Atembeh Noura, Cromwel Zemnou-Tepap, Dowbiss Meta-Djomsi, Martin Maïdadi-Foudi, Ginette Irma Kame-Ngasse, Inès Nyebe, Larissa Gertrude Djukouo, Landry Kengne Gounmadje, Dimitri Tchami Ngongang, Martin Gael Oyono, Cynthia Paola Demeni Emoh, Hervé Raoul Tazokong, Gadji Mahamat, Cyprien Kengne-Ndé, Serge Alain Sadeuh-Mba, Ndongo Dia, Giuseppina La Rosa, Lucy Ndip, Richard Njouom.

**Project administration:** Sebastien Kenmoe, Richard Njouom.

**Supervision:** Sebastien Kenmoe.

**Validation:** Sebastien Kenmoe, Jean Thierry Ebogo-Belobo, Donatien Serge Mbaga, Arnol Bowo-Ngandji, Joseph Rodrigue Foe-Essomba, Serges Tchatchouang, Marie Amougou Atsama, Jacqueline Félicité Yénigué, Raoul Kenfack-Momo, Alfloditte Flore Feudjio, Alex Durand Nka, Chris Andre Mbongue Mikangue, Jean Bosco Taya-Fokou, Jeannette Nina Magoudjou-Pekam, Efietngab Atembeh Noura, Cromwel Zemnou-Tepap, Dowbiss Meta-Djomsi, Martin Maïdadi-Foudi, Ginette Irma Kame-Ngasse, Inès Nyebe, Larissa Gertrude Djukouo, Landry Kengne Gounmadje, Dimitri Tchami Ngongang, Martin Gael Oyono, Cynthia Paola Demeni Emoh, Hervé Raoul Tazokong, Gadji Mahamat, Cyprien Kengne-Ndé, Serge Alain Sadeuh-Mba, Ndongo Dia, Giuseppina La Rosa, Lucy Ndip, Richard Njouom.

**Writing – original draft:** Amary Fall, Sebastien Kenmoe.

**Writing – review & editing:** Amary Fall, Sebastien Kenmoe, Jean Thierry Ebogo-Belobo, Donatien Serge Mbaga, Arnol Bowo-Ngandji, Joseph Rodrigue Foe-Essomba, Serges Tchatchouang, Marie Amougou Atsama, Jacqueline Félicité Yénigué, Raoul Kenfack-Momo, Alfloditte Flore Feudjio, Alex Durand Nka, Chris Andre Mbongue Mikangue, Jean Bosco Taya-Fokou, Jeannette Nina Magoudjou-Pekam, Efietngab Atembeh Noura, Cromwel Zemnou-Tepap, Dowbiss Meta-Djomsi, Martin Maïdadi-Foudi, Ginette Irma Kame-Ngasse, Inès Nyebe, Larissa Gertrude Djukouo, Landry Kengne Gounmadje, Dimitri Tchami Ngongang, Martin Gael Oyono, Cynthia Paola Demeni Emoh, Hervé Raoul Tazokong, Gadji Mahamat, Cyprien Kengne-Ndé, Serge Alain Sadeuh-Mba, Ndongo Dia, Giuseppina La Rosa, Lucy Ndip, Richard Njouom.

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
