## [Decision Letter · Decision Letter 0]

5 Oct 2021

Dear PhD Njouom,

Thank you very much for submitting your manuscript "Global prevalence and case fatality rate of Enterovirus D68 infections, a systematic review and meta-analysis" for consideration at PLOS Neglected Tropical Diseases. As with all papers reviewed by the journal, your manuscript was reviewed by members of the editorial board and by several independent reviewers. In light of the reviews (below this email), we would like to invite the resubmission of a significantly-revised version that takes into account the reviewers' comments. 

We cannot make any decision about publication until we have seen the revised manuscript and your response to the reviewers' comments. Your revised manuscript is also likely to be sent to reviewers for further evaluation.

Sincerely,

Anne W. Rimoin

Associate Editor

Jen-Ren Wang

Deputy Editor

Reviewer's Responses to Questions

**Key Review Criteria Required for Acceptance?**

**Methods**

-Are the objectives of the study clearly articulated with a clear testable hypothesis stated?

-Is the study design appropriate to address the stated objectives?

-Is the population clearly described and appropriate for the hypothesis being tested?

-Is the sample size sufficient to ensure adequate power to address the hypothesis being tested?

-Were correct statistical analysis used to support conclusions?

-Are there concerns about ethical or regulatory requirements being met?

Reviewer #1: YES

Reviewer #2: Methods section does not include the description of how the data acquasition, database mining and data analysis have been performed.

**Results**

-Does the analysis presented match the analysis plan?

-Are the results clearly and completely presented?

-Are the figures (Tables, Images) of sufficient quality for clarity?

Reviewer #1: YES

Reviewer #2: The Results are presented scarcely - a lot of analysis is missing, but is discussed aferwards.

Table titles, column headings and figure captions are badly done and missing the details on data presented.

**Conclusions**

-Are the conclusions supported by the data presented?

-Are the limitations of analysis clearly described?

-Do the authors discuss how these data can be helpful to advance our understanding of the topic under study?

-Is public health relevance addressed?

Reviewer #1: YES

Reviewer #2: It seems that the conclusions are supported by the presented data (in supplementary tables) and the discussion is sufficient.

However, I cannot evaluate the conclusions until the results are altered.

**Editorial and Data Presentation Modifications?**

Reviewer #1: NO

Reviewer #2: (No Response)

**Summary and General Comments**

Reviewer #1: 1) it is better to provide a graph showing the EV-D68 cases over the years. 

2) The following general reviews of EV-D68 virology and antiviral drug discovery should be added:

Elrick MJ, Pekosz A, Duggal P. Enterovirus D68 molecular and cellular biology and pathogenesis. J Biol Chem. 2021 Jan-Jun;296:100317. doi: 10.1016/j.jbc.2021.100317. Epub 2021 Jan 21. Erratum in: J Biol Chem. 2021 Jan-Jun;296:100587. PMID: 33484714; PMCID: PMC7949111.

Hu Y, Musharrafieh R, Zheng M, Wang J. Enterovirus D68 Antivirals: Past, Present, and Future. ACS Infect Dis. 2020 Jul 10;6(7):1572-1586. doi: 10.1021/acsinfecdis.0c00120. Epub 2020 May 14. PMID: 32352280; PMCID: PMC8055446.

3) The authors might want to add the discussion of the comparison between historic and contemporary EV-D68 strains.

Reviewer #2: The study is interesting and long needed. The EV-D68 infection remains a mystery, the data is scarce. And we still not consider this infection of public importance despite the 2014 outbreak and registration of acute flaccid myelitis cases 'of unknown etiology' annualy. However, the data in the manuscript is not presented clearly that undermines the whole study.

General comments:

The most important data is presented in Tables S6 and S8 and scarcly discussed in the results, if at all. In my point of view, the age distribution, prevalence by year and country are important, they change with time. And presenting it as current infections and past infections without years and ages describes nothing.

For example, Figure 4 presents global prevalence of EV-D68 infections - but when? in 2020? 2010-2020? estimated? how it was estimated?

It seems as the results present some calculated numbers that cannot be checked from the data provided.

Specific comments:

In most of the cases throughout the manuscript the terms 'EV-D68' and 'EV-D68 infection' are used as one. It should be corrected.

Figure captions should be put at the end of the manuscript.

Most of the figures and tables miss dimensions of the quantities: for examples, rates are expressed as cases per 1000? 100000?

Acronyms should be used throughout the manuscript, if used.

Lines 93 and 105 - EV-D68 infection issue is important, but it is not the "major public health concern" or "the cause of significant mortality worldwide". It is interesting and important issue as EV-D68 is rapidly evolving EV able to cause CNS damage. You do not need to add importance to this issue.

Line 100 - EVs are divided in TYPES, and each type can be divided into genotypes. Do not mess up the terms.

Lines 138 and 145 - the investigator names are not important, the procedures are. Please, add the descriptions of the procedures: how the databases were mined, scripts you used (if any), how the data was exctracted from manuscripts, etc.

Lines 170 and 177 - aggregated means divided, probably?

Line 202 - what kind of samples (swabs, lavages)?

Lines 234-242 - did the authors of the analysed papers presumed the AFM to be caused by EV-D68? Did they observe the increase in Ab titers during the AFM course?

Lines 246-252 - I am not sure it should be a part of the results section

Line 270 - I could not find children under 5 years of age in Table S8

Lines 365-367 - where in results this data is presented?

PLOS authors have the option to publish the peer review history of their article (what does this mean?). If published, this will include your full peer review and any attached files.

Reviewer #1: No

Reviewer #2: No
---

## [Decision Letter · Decision Letter 1]

23 Nov 2021

Dear PhD Njouom,

Thank you very much for submitting your manuscript "Global prevalence and case fatality rate of Enterovirus D68 infections, a systematic review and meta-analysis" for consideration at PLOS Neglected Tropical Diseases. As with all papers reviewed by the journal, your manuscript was reviewed by members of the editorial board and by several independent reviewers. The reviewers appreciated the attention to an important topic. Based on the reviews, we are likely to accept this manuscript for publication, providing that you modify the manuscript according to the review recommendations. 

Sincerely,

Anne W. Rimoin

Associate Editor

Jen-Ren Wang

Deputy Editor

Reviewer's Responses to Questions

**Key Review Criteria Required for Acceptance?**

**Methods**

-Are the objectives of the study clearly articulated with a clear testable hypothesis stated?

-Is the study design appropriate to address the stated objectives?

-Is the population clearly described and appropriate for the hypothesis being tested?

-Is the sample size sufficient to ensure adequate power to address the hypothesis being tested?

-Were correct statistical analysis used to support conclusions?

-Are there concerns about ethical or regulatory requirements being met?

Reviewer #1: YES

**Results**

-Does the analysis presented match the analysis plan?

-Are the results clearly and completely presented?

-Are the figures (Tables, Images) of sufficient quality for clarity?

Reviewer #1: YES

**Conclusions**

-Are the conclusions supported by the data presented?

-Are the limitations of analysis clearly described?

-Do the authors discuss how these data can be helpful to advance our understanding of the topic under study?

-Is public health relevance addressed?

Reviewer #1: YES

**Editorial and Data Presentation Modifications?**

Reviewer #1: (No Response)

**Summary and General Comments**

Reviewer #1: The references suggested by the previously round of review were not updated during revision. 

The following general reviews of EV-D68 virology and antiviral drug discovery should be added: Elrick MJ, Pekosz A, Duggal P. Enterovirus D68 molecular and cellular biology and pathogenesis. J Biol Chem. 2021 Jan-Jun;296:100317. doi: 10.1016/j.jbc.2021.100317. Epub 2021 Jan 21. Erratum in: J Biol Chem. 2021 Jan-Jun;296:100587. PMID: 33484714; PMCID: PMC7949111. Hu Y, Musharrafieh R, Zheng M, Wang J. Enterovirus D68 Antivirals: Past, Present, and Future. ACS Infect Dis. 2020 Jul 10;6(7):1572-1586. doi: 10.1021/acsinfecdis.0c00120. Epub 2020 May 14. PMID: 32352280; PMCID: PMC8055446. 3) The authors might want to add the discussion of the comparison between historic and contemporary EV-D68 strains.

PLOS authors have the option to publish the peer review history of their article (what does this mean?). If published, this will include your full peer review and any attached files.

Reviewer #1: No

Figure Files:

Data Requirements:

Reproducibility:

References

---

## [Editor Report · Decision Letter 2]

8 Dec 2021

Dear PhD Njouom,

We are pleased to inform you that your manuscript 'Global prevalence and case fatality rate of Enterovirus D68 infections, a systematic review and meta-analysis' has been provisionally accepted for publication in PLOS Neglected Tropical Diseases.

Best regards,

Anne W. Rimoin

Associate Editor

Jen-Ren Wang

Deputy Editor

---

## [Editor Report · Acceptance letter]

11 Jan 2022

Dear PhD Njouom,

We are delighted to inform you that your manuscript, "Global prevalence and case fatality rate of Enterovirus D68 infections, a systematic review and meta-analysis," has been formally accepted for publication in PLOS Neglected Tropical Diseases.

Best regards,

Shaden Kamhawi

co-Editor-in-Chief

Paul Brindley

co-Editor-in-Chief
